# Validation of the Rosenberg Self-Esteem Scale in Military Personnel of the Spanish Army (RSES-JGA)

**DOI:** 10.3390/healthcare12222301

**Published:** 2024-11-18

**Authors:** José Gabriel Soriano-Sánchez, Sylvia Sastre-Riba

**Affiliations:** 1Department of Science Didactics, Faculty of Humanities and Education Sciences, University of Jaén, 23071 Jaén, Spain; gsoriano@ujaen.es; 2Department of Educational Sciences, Faculty of Letters and Education, University of La Rioja, Street Luis Ulloa, 2, 26004 Logroño, Spain

**Keywords:** self-esteem, scale, reliability, military, validation

## Abstract

Background: Self-esteem is a psychological term that, since its emergence in 1890 with William James, has been widely studied. Furthermore, this construct has been examined in different areas of psychology, including the clinical field, where its relationship with mental health and low levels of depression has been demonstrated. In the realm of positive psychology, its connection to resilience and life satisfaction has also been established. Evaluating self-esteem in military personnel is essential, as healthy self-esteem can enhance resilience, improve performance, and promote emotional well-being. Objective: To evaluate validity based on internal structure from the perspective of Classical Test Theory (CTT) and Item Response Theory (IRT), to obtain evidence of validity based on the relationship with other variables, and to estimate the reliability of the *Rosenberg Self-Esteem Scale* (RSES). Method: To this end, 584 military personnel from the three ranks of the Spanish Army (officers, non-commissioned officers, and professional soldiers and sailors) were evaluated, of which 511 were men (87.70%) and 72 were women (12.30%), with an age range of 18 to 66 years (*M* = 33.17, *SD* = 7.38). Results: The results indicate that the unidimensional structure of the RSES shows adequate fit indices (CFI = 0.94, TLI = 0.91, SRMR = 0.05), while the difficulty and discrimination parameters exhibited favorable values. Additionally, an acceptable reliability value was found (ɷ = 0.81; α = 0.80). Conclusions: Therefore, it is concluded that the RSES-JGA presents adequate psychometric properties of validity and reliability, making it a useful and relevant instrument for measuring self-esteem in military personnel of the Spanish Army. This will make it possible to measure the level of self-esteem in military personnel, i.e., the set of perceptions, thoughts, evaluations, feelings, and behavioral tendencies directed towards oneself, one’s own way of being, and towards one’s own body and character traits.

## 1. Introduction

Self-esteem is a psychological concept that has been the subject of study since 1890, when William James introduced it. The most common definition is that of Rosenberg (RSES) [1], who describes it as the attitude a person has towards themselves, whether positive or negative. Self-esteem, according to Rosenberg in 1965, is measured from a single dimension because it is considered a unidimensional construct that reflects the overall evaluation a person makes of themselves. In his self-esteem scale, Rosenberg argues that this evaluation encompasses both positive and negative aspects of self-image, allowing for the essence of self-concept to be captured in a single measure. This simplification facilitates comparison between individuals and groups, as well as providing an effective tool for investigating how self-esteem influences various behaviors and psychological outcomes. However, the measurement of self-esteem has sparked intense debate in the psychological literature, with authors advocating for both a unidimensional approach and a bidimensional model. Proponents of unidimensional measurement, such as Rosenberg [1], argue that this simplified approach not only facilitates comparison between individuals and groups but also allows for the investigation of how self-esteem influences various behaviors and psychological outcomes [2]. In contrast, advocates of the bidimensional model argue that it is essential to break down self-esteem into separate dimensions, such as self-acceptance and positive self-concept [3]. This approach acknowledges the complexities of human self-perception and suggests that individuals may experience different levels of satisfaction in various areas of their lives. By doing so, it provides a more nuanced understanding of how self-esteem manifests in specific contexts [4]. Nevertheless, this bidimensional perspective also presents significant challenges that must be considered when researching and applying concepts related to self-esteem. Therefore, while both approaches have their merits, using a unidimensional model may be more suitable for obtaining an overall and comparative view of self-esteem across diverse populations and situations [5].

Self-esteem has been researched in various areas of psychology. In the clinical field, it has been found to be related to mental health and lower levels of depression [6]. In positive psychology, it has been linked to resilience and life satisfaction [7]. In the military context, an adequate level of self-esteem is crucial for combatants to successfully face survival situations [8]. Burger and Bachmann [9] highlight that living in hostile environments characterized by violence can harm self-esteem, while a warm and safe environment strengthens it. Developing social skills in military personnel could be beneficial, as these are closely related to self-esteem and physical activity [10], which could enhance their professional effectiveness and personal well-being. Self-esteem also acts as a protective factor against stress in combat; military personnel with high self-esteem tend to use healthy strategies to manage stress [11] and have self-confidence [12]. A study by Lee et al. [13] examined the relationship between post-traumatic stress disorder (PTSD), stress, depression, self-esteem, and impulsivity. The results showed that PTSD is positively related to depression, stress, and impulsivity, while it has a negative relationship with self-esteem. In turn, self-esteem is closely related to resilience, as a positive attitude towards oneself facilitates adaptation to the environment, which in turn increases an individual’s ability to cope with pressures in various situations [14]. Thus, adequate self-esteem acts as a protective factor that enhances effectiveness in managing such circumstances [15].

Currently, several instruments have been developed to measure self-esteem. Some of the most notable are the *Coopersmith Self-Esteem Inventory* [16], the *Collective Self-Esteem Scale* by Diener et al. [17], and the *Single-Item Self-Esteem Scale* [18]. However, the most widely used scale is the RSES [1], which consists of 10 items that assess positive or negative self-evaluation. In Latin America, adaptations of this scale have primarily focused on adolescents [15], with few studies directed at adults. In contrast, in some European countries, such as Spain, validation studies focused on adults have been conducted. For example, Martín-Albo et al. [19] validated the Rosenberg scale in 2007 specifically for university students. Similarly, Mayordomo et al. [20] carried out an adaptation and validation of this scale with the aim of evaluating the structure of Rosenberg’s unifactorial model in a sample of adults over seventy years old. In contrast, there is no instrument that has verified the reliability and validity of the *Rosenberg Self-Esteem Scale* [1] in military personnel of the Spanish Army.

Nevertheless, validation studies have presented methodological limitations that raise doubts about their results. In exploratory factor analysis (EFA), many studies [21,22,23] used the Little Jiffy method, which does not account for measurement error, potentially leading to an overestimation of factor loadings and explained variance [24]. Additionally, reliability is estimated using Cronbach’s alpha and McDonald’s omega. Alpha requires meeting assumptions such as tau-equivalence and unidimensionality of the instrument [25], as well as continuous variables with interval measurement [26]. On the other hand, omega requires factor loadings obtained from a confirmatory factor analysis [27].

Finally, the factor structure of the RSES is a subject of debate, with positions considering it to be either bidimensional or unidimensional. A meta-analysis suggests a two-factor structure, composed of positive and negative items [28], which has also been supported by confirmatory factor analyses [29]. The negative wording of some items may contribute to the emergence of a new dimension [30]. Additionally, a method effect associated with negative items has been identified [31]. However, this effect persists in the RSES despite the efforts made to address it.

### The Present Study

Based on the above, the present study aims to evaluate the validity based on internal structure from the perspective of Classical Test Theory (CTT) and Item Response Theory (IRT) in order to obtain evidence of validity related to other variables and to estimate the reliability of the RSES in military personnel of the Spanish Army. Based on previous scientific evidence, the following research hypothesis (H) is proposed:

**H1.** 
*Self-esteem is positively related to resilience in military personnel of the Spanish Army.*


## 2. Materials and Methods

### 2.1. Participants, Procedures, and Instruments

The sample consisted of 584 soldiers from the Spanish Army (4.5% officers, *n* = 27; 17.5% non-commissioned officers, *n* = 102; and 78%, *n* = 455 professional troops and sailors), with an average age of 33.17 years (*SD* = 7.38), ranging from 18 to 66 years. Regarding gender, 87.70% (*n* = 512) were men, while 12.30% (*n* = 72) were women. Specifically, the sample belonged to the same barracks, composed of both operational units and support units. The study sample was extracted through non-probabilistic intentional sampling.

The present study has a quantitative approach and presents an instrumental design, as the objectives are oriented towards evaluating the psychometric properties of a measurement instrument [32].

The study was approved by the Ethics Committee of the Central Defense Hospital (Approval Code: 51117). Once this permission was obtained, the purpose of the study was communicated to the military authorities stationed at the King Alfonso XIII Legion Brigade (BRLIEG, Almería, Spain), specifically to the Honorable Colonel Chief of the Alvarez de Sotomayor Services Unit (Chief of the Barracks) and to the Excellency General (Chief of BRILEG). Subsequently, after receiving authorization for the research from these officials, several meetings were held with leaders of different units to inform them about the objectives of the research, ensuring them that data confidentiality would be maintained. During these meetings, a work schedule was developed indicating the day each unit (both operational and support units) was to participate. In this regard, prior to participation in the study, participants were informed about its objective and voluntary nature, both verbally and in writing, outlining the procedure for participation, anonymous treatment, and data confidentiality, adhering to ethical research standards as established by the Helsinki Declaration [33], which guarantees the following criteria: (a) confidentiality of collected data and their exclusive use for research purposes; (b) anonymity of data; (c) professional secrecy in data collection.

After being informed, each participant was then given a questionnaire individually, which was placed in an envelope that they were required to return sealed in its respective envelope after completing it. This process took an average time of 25–30 min.

Regarding the instruments used, first an ad hoc questionnaire was used to collect sociodemographic data (age and gender). First, the authors developed a booklet containing already validated instruments along with an ad hoc questionnaire that collected sociodemographic aspects of the participants (gender and age). To assess self-esteem, the RSES [1] was used. This instrument is made up of 10 items with 4 response options (strongly agree: 4; agree: 3; disagree: 2; strongly disagree: 1). This allows for a score ranging from 10 (minimum self-esteem) to 40 (maximum). Examples of the items that make up the scale would be: 1. I feel that I am as valuable a person as others; or 2. In general, I am inclined to think of myself as a failure. The reliability and validity of the scale are good, as reported in various studies such as that of Martín-Albo et al. [19] with α = 0.85, the one by Rojas-Barahona et al. [34] with α = 0.75, and even the one provided by the original author of the scale [1] with α = 0.75. This study presents an *α* = 0.80 (ɷ = 0.81). On the other hand, for resilience, the *Resilience Scale* (RS) developed by Wagnild and Young [35] was used. It is one of the few reliable and valid psychometric tools for assessing levels of psychosocial adaptation to significant life events [36]. The RS consists of 25 items with a *Likert* response format that ranges from 1 (strongly disagree) to 7 (strongly agree). The scoring ranges from 25 to 175 points, such that: above 150 indicates optimal resilience, between 166 and 150 points indicates high values, between 101 and 125 points indicates medium values, and below 100 points indicates low resilience. Examples of the items that make up the scale would be: 1. When I plan something, I do it; or 2. I usually manage one way or another. According to Wagnild and Young [35], the items of the scale have the following characteristics of resilience: 1. Perseverance. This refers to the desire to keep fighting to fulfill the life cycle, persisting through adverse situations that involve the individual in their life. The person must remain engaged in their proper development and maintain discipline across various contexts that encompass them. 2. Equanimity. This involves the ability to remain calm in a wide range of situations, approaching these situations with composure and maintaining appropriate responses in adverse circumstances. 3. Personal satisfaction. This refers to the evaluation of one’s own contributions, understanding that life has a singular meaning. 4. Being happy alone. This encompasses the idea that each person’s life is unique and that some experiences are faced alone while still feeling good about it. When one feels good alone, they possess the meaning of being unique, as well as a sense of freedom. 5. Self-confidence. This consists of an individual’s ability to rely on themselves while recognizing their own limitations and strengths. In this sense, these five characteristics of the RS are evaluated from the following factors: Factor 1. Personal competence. It includes the level of self-confidence, independence, invincibility, power, ingenuity, and perseverance; Factor 2. Acceptance of oneself and life. It includes the level of flexibility, balance, and perspective of life that coincides with acceptance of life and a feeling of peace despite adversity. In terms of reliability, studies such as the one conducted by Dias et al. [37] report an alpha coefficient of *α* = 0.72. Meanwhile, Wagnild and Young [35] found even greater reliability, with an *α* = 0.90. In this study, the SR presents an *α* = 0.92 (ω = 0.93). Regarding the factors, Factor 1 “Personal competence” presents a reliability of *α* = 0.93 (ω = 0.94) and Factor 2 “Acceptance of oneself and life” of *α* = 0.91 (ω = 0.92).

Consequently, analyzing how self-esteem positively relates to resilience in soldiers of the Spanish Army is essential because high self-esteem can strengthen military personnel’s ability to adapt and recover from difficult situations, which is crucial in their work environment. Furthermore, this relationship can provide valuable insights for the validation of the scale, ensuring that assessment tools are effective and relevant in addressing the psychological and emotional needs of military personnel, thereby promoting their well-being and effectiveness in service.

### 2.2. Data Analysis

First, the database was cleaned by removing outlier data. Next, descriptive analyses of mean (*M*), standard deviation (*SD*), as well as skewness (As) and kurtosis (Ku) were estimated, with values considered acceptable when As < ±2 and Ku < ±7 [38]. On the other hand, Pearson correlation analyses were performed to explore the relationship between the variables. The value of *Pearson’s r* coefficient is between (−1) and (+1). For its interpretation, it is considered: *r* = 0 (no association or correlation between the two variables); 0 < *r* < 0.25 (weak correlation); 0.25 ≤ *r* < 0.75 (intermediate correlation); 0.75 ≤ *r* < 1 (strong correlation); and *r* = 1 (perfect correlation). Subsequently, confirmatory factor analysis (CFA) was conducted using the Diagonally Weighted Least Squares with Mean and Variance corrected (WLSMV) estimator, as the data were ordinal [39]. For model evaluation, the fit indices used were chi-square (*χ*^2^), Comparative Fit Index (CFI), Tucker–Lewis Index (TLI), Root Mean Square Error of Approximation (RMSEA), and Standardized Root Mean Square Residual (SRMR). Cut-off points for evaluating the fit indices were considered based on Hu and Bentler’s proposal: CFI > 0.95, TLI > 0.95, RMSEA < 0.08, SRMR < 0.08 [40]. Regarding validity related to other variables, the structural equation modeling (SEM) approach was employed to evaluate the relationship between self-esteem and resilience. For this purpose, the WLSMV estimator was used due to the ordinal nature of the data, and the evaluation was conducted using the same values as in the CFA. For Item Response Theory (IRT), a Graded Response Model (GRM) [41] was utilized, specifically an extension of the two-parameter logistic model (2-PLM) for ordered polytomous items [42]. For each item, two types of parameters were estimated: (a) discrimination and (b) difficulty. The discrimination parameter determines the slope at which responses to the items change based on the level of the latent trait, while the item difficulty parameters determine how much of the latent trait is required for an item to be answered.

On the other hand, for the comparison of means with polytomous predictor variables (with three or more values) and criterion variables, the ANOVA test was used for the analysis of variance. For this, the predictor variable was the military scale to which the student belongs, and the criterion variable was self-esteem. The classification levels for the effect size are those indicated in the eta squared values (*η*^2^), specifically: *η*^2^ = 0.01 (small); *η*^2^ = 0.06 (medium); and *η*^2^ = 0.14 (large). The Bonferroni test was applied for post hoc contrasts. Likewise, to find out if there is a statistical difference between means, gender and its relationship with the criterion variable (self-esteem) were used as a dichotomous predictor variable, through the analysis of the Student’s *t* test for independent samples, assessing the effect by means of the difference in means and the precision by means of the 95% confidence interval. The effect size was calculated using Cohen’s d statistics to check the difference in means in the dispersion of two samples. A *d* = 0.2 or less is considered a small effect size; a *d* = 0.5 is considered a medium effect size; and a *d* = 0.8 or greater is considered a large effect size. Descriptive data and reliability were calculated using SPSS statistical software version 25.0 [43], and confirmatory factor analyses were performed with AMOS software version 24.0 [44].

### 2.3. Revision, Translation, and Adaptation to the Military Context

Since the instrument was created for a different population, a process of translation, adaptation, and standardization was necessary to achieve content validity. The items were first translated into Spanish by a native translator (back-translation), considering four criteria [45]: 1. The cultural context in which the adaptation will take place; 2. Technical aspects of the development and adaptation of the test; 3. Test administration; 4. Interpretation of the scores.

Subsequently, a military member from one of the participating units at the Almeria garrison verified the items to ensure that the questions were relevant to military personnel.

### 2.4. Content Validation

For content validity, expert judgment was used, which is a useful validation method for verifying the reliability of a survey [46]. Of the total number of experts, two were from the University of La Rioja and were specialists in testing and psychometrics, while the other two were military personnel belonging to the officer corps, with more than ten years of experience.

## 3. Results

Firstly, in Table 1, it can be seen that the average score of the 10 items ranges from 2.91 (*SD* = 1.37) to 3.59 (*SD* = 0.69). Additionally, the skewness and kurtosis indicate that all items have adequate values (As < ±2; Ku < ±7), according to the criteria set by Finney and DiStefano [38].

Regarding the correlation between items, a statistically significant correlation was found between all of them (*p* < 0.01) (Table 2).

### 3.1. Evidence Based on Internal Structure

Regarding the factorial structure with a general factor, the confirmatory factor analysis demonstrated fit indices (*χ*^2^ = 118.18, df = 26, *p* < 0.001, CFI = 0.94, TLI = 0.91, RMSEA = 0.07, 90% CI [0.064, 0.093], SRMR = 0.05), and the factor loadings of the items were above 0.95 (Figure 1).

### 3.2. Validity Based on the Relationship with Other Variables

To obtain evidence of validity related to other variables, a model was developed using the SEM approach to evaluate the relationships between the RSES and the variable resilience. The results show that the proposed model demonstrates adequate values in the fit indices (*χ*^2^ = 1202.36, df = 525, *p* < 0.001, CFI = 0.92, TLI = 0.91, RMSEA = 0.047, 90% CI [0.44, 0.51], SRMR = 0.05). Additionally, the items showed appropriate values in representing the measurement variables. Regarding relationship values, positive correlations were found between self-esteem and global resilience (*r* = 0.30; *p* < 0.01), as well as between self-esteem and Factor 1 of resilience “Personal competence” (*r* = 0.32; *p* < 0.01) and Factor 2 “Acceptance of oneself and life” (*r* = 0.30; *p* < 0.01).

### 3.3. Item Response Theory Model: Graded Response Model

The results found in the confirmatory factor analysis support the two main assumptions: the existence of unidimensionality and, consequently, local independence. Therefore, a graded response model (GRM) was used, specifically an extension of the two-parameter logistic model (2-PLM) for ordered polytomous items. Table 3 shows that all items have discrimination parameters above the value of 1, which is generally considered good discrimination [36]. Regarding the difficulty parameters, all threshold estimates increased monotonically. That is, a greater presence of the latent trait is required to respond to the higher response categories.

### 3.4. Analysis of Variance, t-Test, and Correlation

The ANOVA and Bonferroni post hoc analysis of variance carried out to determine the differences between self-esteem according to the military scale predictor variable reported that there were no intragroup differences in self-esteem (Table 4).

On the other hand, the results obtained in the Student’s *t*-test allowed us to observe that there were no differences in means in self-esteem with respect to gender (Table 5).

The correlation analysis indicated that no correlation was found between self-esteem and age in the military personnel of the Spanish Army (r = 0.41; *p* = 0.27).

### 3.5. Reliability

In the study, the scale demonstrates adequate indices of internal consistency (ɷ = 0.81; *α* = 0.80). Finally, the structure of the self-esteem scale items is presented in this study, as shown in the following (see Appendix A).

## 4. Discussion

The present research aimed to evaluate the validity based on internal structure from the perspective of CTT and IRT, in order to obtain evidence of validity based on the relationship with other variables and to estimate the reliability of the RSES in military personnel of the Spanish Army. Regarding the internal structure of the model, it is evident that most fit indices are adequate for the unidimensional model (CFI = 0.94, TLI = 0.91, RMSEA = 0.07, SRMR = 0.05). These results align with the study by Vilca et al. [47], which reported a unidimensional structure in another line of research. The results are similar to those found in previous studies that identify a good fit for the unidimensional model [19,48,49]. On the other hand, the results found in the reliability analysis reported an adequate value in Cronbach’s alpha coefficient (α = 0.80), similar to that found in previous studies with university students [23,49,50,51]. Additionally, an adequate value was evidenced in the omega coefficient (ɷ = 0.81). This result is also similar to those found in other studies [52,53].

In relation to the research hypothesis, our results have shown that self-esteem is related to resilience [7]; therefore, this research hypothesis is accepted. In this regard, it could be inferred that the development of resilience or self-esteem positively impacts the military personnel of the Spanish Army. Therefore, if self-esteem is developed in this context, understood as the valuation and perception that a person has of themselves [1], it would also foster their level of resilience, which constitutes the capacity that allows a person to face unpleasant situations, grow from them, and prevent them from being affecting intrinsically [54]. In fact, in the Army, resilience has traditionally been considered in terms of morale; it has always been a fundamental issue [55], as it was understood that success or failure in operations depended on it, given that its members experience significant psychological, physical, relational, and social wear and tear [56]. However, positive psychological resources, such as resilience, have a buffering effect on self-esteem, making psychopathological components like anxiety and depression, phobias, or psychoticism less impactful [57].

Authors such as Diener [17] explain that individuals with a positive self-perception have greater confidence in pursuing their current and future projects, resulting in a more meaningful life. Additionally, this result is similar to those found in other studies [58,59]. Furthermore, the relationship between self-esteem and depression, anxiety, and stress [60] can be explained from the theoretical perspective of self-esteem, as developing high self-esteem leads to better self-acceptance, self-confidence, and self-efficacy, which are protective factors that reduce levels of depression, anxiety, and stress that an individual may experience. There are also various studies supporting the positive influence of self-esteem on these variables [1,61,62].

Regarding the IRT models, optimal values were found in the discrimination parameters. Such findings indicate that these items are more precise indicators for evaluating the construct. This allows respondents to have greater ease in differentiating their choice of response alternatives based on the presence of the latent trait. Concerning the difficulty parameter, it was found that a greater presence of the latent trait is necessary to choose the higher response categories. Additionally, regarding the TIC, it was identified that the scale is useful and reliable for recognizing university students with low levels of self-esteem. On the other hand, our results indicate that there are no differences in self-esteem levels based on the soldier’s gender or military rank, and that self-esteem is not related to the age of military personnel. This suggests that both men and women, as well as soldiers from the Troop and Navy Scale, non-commissioned officers, and officers, report similar levels of self-esteem. Therefore, operational activities, such as those in the Troop and Navy Scale, do not seem to affect their self-esteem. Additionally, military personnel, regardless of their age, can have adequate levels of self-esteem, as the results suggest that self-esteem does not develop with increasing age in the Spanish Army.

Although the *Rosenberg Self-Esteem Scale* of 1965 had already been validated, this study confirms its suitability for the Spanish context. A translation into Spanish was carried out by a Committee of Experts, highlighting that, while previous research such as that by Martín-Albo [19] had adapted it for university populations, it is essential to translate it for the military population, as members of the Spanish Army face specific challenges, such as operational stress and discipline, which can influence their self-esteem. Therefore, a cultural and linguistic adaptation could ensure that the items are relevant and understandable for military personnel, allowing for more accurate data collection. The results obtained show adequate validity and reliability indices in this unidimensional scale, facilitating precise measurement of the positive self-evaluation of military personnel. Self-esteem is a fundamental psychological construct that impacts emotional and social well-being, as well as professional performance. This scale establishes a solid foundation for investigating the relationship between self-esteem and other psychological factors, such as resilience. Regarding the unidimensionality of the scale, this characteristic may be due to all items being oriented to capture a single dimension of self-concept, that is, how a person feels generally about their worth and personal dignity. Although self-esteem can manifest in different contexts (social, academic, professional), the scale aims to provide a comprehensive and coherent measure of this construct, allowing for the assessment of overall self-esteem based on an individual’s general perception of themselves, encompassing both positive and negative aspects of self-evaluation.

However, despite the results obtained, this study reveals some limitations. First, a purposive sampling was conducted, which does not allow for the generalization of the results to other contexts within the Army, such as the Navy or the Air Force. Additionally, reliability was not assessed through a test–retest method, which would help identify whether there are variations in scores over time. Therefore, future research on this scale should evaluate the stability over time of the scores through test–retest reliability and its possible modification through clinical treatment. Likewise, it would be advisable to evaluate the psychometric properties of the scale in military personnel of the Navy and the Air Force. Finally, this study demonstrates that the new self-esteem scale, called the *Rosenberg-Jose Gabriel Army Self-Esteem Scale* (RSES-JGA), exhibits adequate psychometric properties. Although this scale is not new, the present study has confirmed the reliability and validity of the scale for optimally measuring the level of self-esteem in military personnel of the Spanish Army. Additionally, support was provided for the unidimensional structure from both a CTT and IRT perspective, and it was reported that this new scale has validity evidence based on its relationship with resilience, a relevant variable for successfully coping with adverse situations throughout a military career. Furthermore, it shows an adequate reliability value.

## 5. Conclusions

The RSES-JGA demonstrates adequate psychometric properties of validity and reliability in military personnel of the Spanish Army. This provides a useful and relevant measure for assessing self-evaluation in the military context.

The unidimensional evaluation of the RSES-JGA scale presents as a strength its brevity and easy administration, which are extremely useful in evaluating the level of self-esteem in the military personnel of the Spanish Army, with the purpose of promoting their well-being and quality of life. The results of the present study support the usefulness of the validation of the RSES-JGA scale as an instrument that shows evidence of reliability and validity to measure the level of self-esteem in the military context of the Spanish Army. This instrument could contribute to a better understanding to analyze the level of self-esteem of military personnel.

In sum, the present instrument will reliably and validly measure the level of self-esteem in military personnel; that is, the set of perceptions, thoughts, evaluations, feelings, and behavioral tendencies directed towards oneself, towards one’s own way of being, and towards the features of one’s own body and character.

## Figures and Tables

**Figure 1 healthcare-12-02301-f001:**
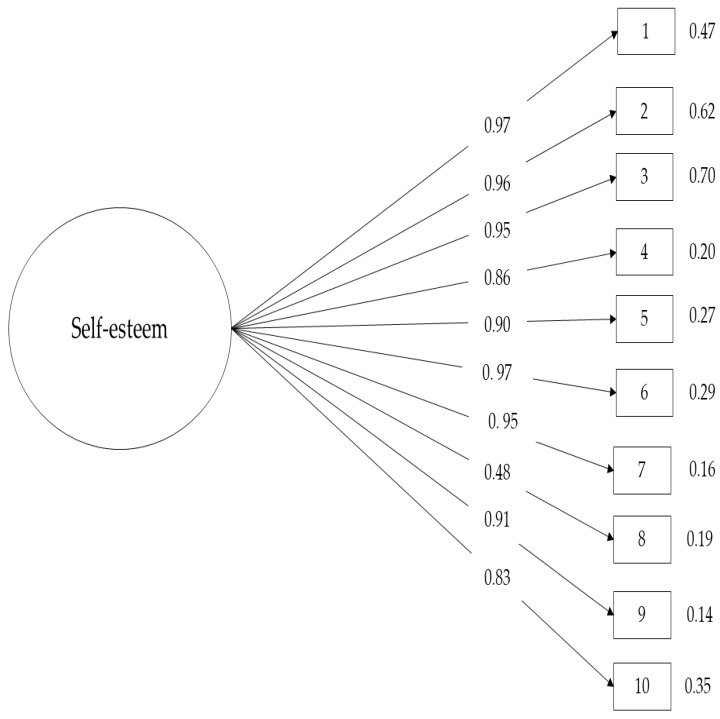
Confirmatory factor analysis of the Rosenberg Self-Esteem Scale.

**Table 1 healthcare-12-02301-t001:** Descriptive analysis of the items and correlation matrix.

Item	*M*	*SD*	g1	g2
A1	3.33	0.82	1.24	1.09
A2	3.41	0.89	1.03	0.64
A3	3.02	1.00	0.13	1.87
A4	3.45	0.58	0.49	0.69
A5	3.18	1.06	0.96	0.52
A6	3.38	0.69	0.99	1.01
A7	3.37	0.60	0.39	0.67
A8	3.39	0.48	0.46	1.82
A9	2.91	1.37	0.62	1.54
A10	3.59	0.69	1.71	1.28

*Note*. M = Mean; SD = standard deviation; g1 = skewness; g2 = kurtosis.

**Table 2 healthcare-12-02301-t002:** Correlation matrix.

Item	A1	A2	A3	A4	A5	A6	A7	A8	A9	A10
A1	1	0.23 **	0.30 **	0.37 **	0.14 **	0.34 **	0.43 **	0.14 **	0.30 **	0.30 **
A2		1	0.25 **	0.34 **	0.27 **	0.23 **	0.28 **	0.31 **	0.46 **	0.25 **
A3			1	0.54 **	0.16 **	0.40 **	0.43 **	0.25 **	0.31 **	0.22 **
A4				1	0.20 **	0.43 **	0.51 **	0.25 **	0.39 **	0.33 **
A5					1	0.21 **	0.22 **	0.19 **	0.29 **	0.27 **
A6						1	0.54 **	0.28 **	0.37 **	0.32 **
A7							1	0.31 **	0.52 **	0.37 **
A8								1	0.52 **	0.30 **
A9									1	0.52 **
A10										1

*Note*. ** = significant correlation at the 0.01 level (two-tailed).

**Table 3 healthcare-12-02301-t003:** Discrimination and difficulty parameters for the scale items.

Unidimensional Model				
Item	a	b_1_	b_2_	b_3_
1	2.76	−2.07	−1.22	0.56
2	2.54	−2.15	−1.60	0.47
3	2.63	−2.23	−1.06	0.61
4	2.83	−2.25	−1.13	0.63
5	3.72	−2.06	−0.98	0.48
6	3.87	−1.82	−0.85	0.75
7	3.35	−1.81	−0.76	0.83
8	4.18	−2.05	−0.82	0.80
9	2.72	−1.99	−0.93	0.78
10	3.55	−2.23	−1.22	0.56

*Note.* a: discrimination parameters; b: difficulty parameters.

**Table 4 healthcare-12-02301-t004:** Self-esteem: military scale. Results of the analysis of variance.

	Groups	*N*	*M*	*SD*	ANOVA			*Post Hoc* Contrasts
					*F*(3.735)	*p*	*η^2^*	
Military scale	Troop (g1)	455	28.67	3.43	0.18	0.835	0.010	│g1 < g2││g1 > g3││g2 > g3│
	Non-commissioned officers (g2)	102	28.74	3.54				
	Officials (g3)	27	29.34	3.61				

**Table 5 healthcare-12-02301-t005:** Self-esteem by gender; *t*-test results.

		*n*	*M*	*SD*	*t*	*p*	*d*
Self-esteem	Male	512	28.67	3.35	1.72	0.066	0.12
	Female	72	27.73	3.61			

## Data Availability

The data supporting the research can be requested from the corresponding author.

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
