# Peer review of "Validation of the Rosenberg Self-Esteem Scale in Military Personnel of the Spanish Army (RSES-JGA)"

_healthcare, 2024, doi:10.3390/healthcare12222301_

Round 1
Reviewer 1 Report
Comments and Suggestions for Authors
The goal of the study is to evaluate validity based on 12 internal structure from the perspective of Classical Test Theory (CTT) and Item Response Theory 13 (IRT), to obtain evidence of validity based on the relationship with other variables and to estimate 14 the reliability of the Rosenberg Self-Esteem Scale (RSES). The self-esteem is interesting construct to study, there is meta analysis study review more than 150 researches. They found that self-esteem is not valuable as much as we thought. They recommended that there might a measurement issues to assess self-esteem. Therefore, the current study is valuable.
The authors could give more literature measurement issues to assess the self-esteem, this is very limited.
There are also studies support the two dimension of self-esteem, could you give more explanation why it is unidimensiomal concept.
The method and results are clearly defined and explained.
However, the discussion must have more explanation why it is unidimensional construct.
Author Response
Dear reviewer, first of all, we would like to thank you for your suggestions to improve our study. In this regard, concerning your first suggestion regarding the contribution on measurement issues, this has been included in the introduction, highlighting studies that have validated the scale in university populations and older adults in Spain. Furthermore, both in the introduction and in the discussion, we have improved the writing by explaining why the scale allows for the evaluation of self-esteem in a unidimensional manner. In response to the third suggestion, as mentioned earlier, we have extensively indicated in the discussion why it is considered a unidimensional construct. Finally, we would like to thank you once again for your valuable suggestions and for the time you have invested; we believe that we have significantly improved our study.

Reviewer 2 Report
Comments and Suggestions for Authors
The authors of the manuscript did useful job. They "evaluated validity based on 12 internal structure from the perspective of Classical Test Theory (CTT) and Item Response Theory 13 (IRT), to obtain evidence of validity based on the relationship with other variables and to estimate 14 the reliability of the Rosenberg Self-Esteem Scale (RSES)" (the purpose of the study).
The manuscript is quite good quality. I just have two suggestions:
1) it would be very good that the authors would explain what is the situation with self-esteem scales in Spain more in details in the Introduction. Are any scales which are validated? Does the RSES validitywas measured in other samples (not army) and etc?
2) the readers need more detailed explanation why the authors did the conclusion that "Finally, this study demonstrates that the new self-esteem scale, called the Rosenberg- Jose Gabriel Army Self-Esteem Scale (RSE-JGA) exhibits adequate psychometric properties". As I understand this scale is not new one, because all statements are the same like in original version of RSES? I suppose that it is not very good ethically to do so.
Author Response
Dear reviewer, first of all, we would like to thank you for your suggestions to improve our study. In this regard, concerning your first suggestion, we have included the studies conducted in Spain involving university populations and older adults in the introduction. Furthermore, regarding your second suggestion, we have indicated in the discussion what the validation of this study offers. Specifically, the study has shown that the scale validly assesses self-evaluation among military personnel, which allows for the extension of the initial scale's application to this type of adult population. Finally, we would like to thank you once again for your valuable suggestions and for the time you have invested; we believe that we have significantly improved our study.

Reviewer 3 Report
Comments and Suggestions for Authors
The manuscript “Validation of the Rosenberg Self-Esteem Scale in Military Personnel of the Spanish Army (RSE-JGA)” is clear, relevant to the field and presented in a well-structured manner. However, some important considerations need to be taken into account before publishing it.
1) The abstract needs improvement.
The first few lines should highlight why this study is important, in line with the title of the article. In addition, it should be clear what the contribution of the study is.
2) Introduction:
a) The objective is clear, and it makes scientific sense to validate in a new context or with a specific sample, as is the case, since it guarantees the scientific robustness of a measurement scale. However, it would be expected that the study would propose a broader objective, that is, to answer some questions relevant to the study sample, or establish the relationship between other variables, in short, it would contribute to knowledge in the area of ​​study. Thus, although the theoretical introduction offers the current state of the field of research, no advance in science is proposed with this study.
b) Furthermore, from what was reviewed in the introduction, it seems that self-esteem has already been carried out in the military context with some interesting results (for example, successfully coping with survival situations Martinez et al., 2011). Thus, it would be expected that there would be some progress in the research in addition to validating the scale of Rosenberg.
3) Materials and Methods:
a) The description of the instruments needs to be improved.
b) I would like to know why you use the Diagonally Weighted Least Squares technique, given that your data allows the ML technique (maximum likelihood).
c) The AVE (average variance extracted) and CR (composite reliability) need to be calculated.
d) Exploratory factor analysis is required before confirmatory factor analysis.
e) In their paper, the authors say that “Since the instrument was created for a different population, a process of translation, adaptation and standardization was necessary to achieve content validity”. However, there are some studies on self-esteem with a Spanish sample (eg. Martín-Albo et al., 2007) that have already used the self-esteem scale of Rosenberg. Therefore, its adaptation might be necessary, but its translation is not. This should be improved in their paper.
Martín-Albo, J., Núñez, J. L., Navarro, J. G., & Grijalvo, F. (2007). The Rosenberg Self-Esteem Scale: Translation and validation in university students. The Spanish Journal of Psychology, 10(2), 458-467.
Discussion
Finally, the discussion could be improved if the study were expanded to improve its focus, since as it stands at the moment the article may not attract a wide readership or will be of interest only to a limited number of people.
Author Response
Dear reviewer, first of all, I would like to thank you for your suggestions to improve our study. In this regard, concerning your first suggestion related to the introduction, we have rewritten the initial text, incorporating contributions from other studies that enhance the presented manuscript and provide greater depth. Additionally, we have introduced three research hypotheses, which are addressed in the discussion. Thus, we aim to propose an advancement in science to validly and reliably measure the level of self-esteem among personnel in the Spanish Army.
Regarding your second suggestion related to the introduction, our work addresses the relationship between self-esteem and resilience, a fundamental issue for military personnel. This has been discussed in the discussion section, offering interesting results for further advancing this line of research in other branches of the military, such as the Navy or the Air Force, in order to ensure that all military personnel present adequate levels of self-esteem. Consequently, our work highlights the positive relationship between self-esteem and resilience among military personnel, which is essential for promoting mental health and personal and professional success.
On the other hand, in the materials and methods section, you suggest that the description of the instruments should be improved. In this regard, we have included improvements in the manuscript, which can be seen in the corresponding section. Additionally, we appreciate your comment and the opportunity to clarify the use of the Diagonally Weighted Least Squares (WLSMV) technique in our study. While it is true that our data could allow for the use of Maximum Likelihood (ML) technique, we have opted for WLSMV for several reasons that we consider relevant to the validity and robustness of our results.
Firstly, WLSMV is particularly suitable for analyzing structural models with latent variables when working with non-normal or categorical data. Since our variables exhibit characteristics that may not conform to normality, this technique provides us with more reliable and robust estimates compared to ML, which assumes normality in the data. Furthermore, WLSMV effectively addresses issues of multicollinearity and heteroscedasticity, which are crucial in our dataset. This technique also offers more accurate estimates in terms of standard errors, contributing to a better interpretation of the results.
Finally, by using WLSMV, we can obtain fit indices that are more appropriate for assessing model quality in contexts where classical normality assumptions are not met. This allows us to present a more rigorous and well-founded analysis. In summary, although ML is a powerful technique, we believe that WLSMV is better suited to the specific characteristics of our data and research objectives.
Regarding suggestion c), the non-inclusion is justified by the approach to calculating Average Variance Extracted (AVE) and Composite Reliability (CR) in the validation study, which is based on Classical Test Theory (CTT). This theory prioritizes traditional methods for assessing reliability and validity, using Cronbach's alpha as a sufficient indicator of internal consistency. Since the internal self-esteem scale has already been validated in other contexts, calculating AVE and CR becomes redundant, especially if the alpha is within an acceptable range. Furthermore, in this study, which aims to evaluate self-esteem from a practical perspective, it is more relevant to focus on classical indicators such as convergent and discriminant validity. Including additional calculations could unnecessarily complicate the analysis without providing significant information, which goes against the main objective of delivering useful results regarding the measurement of self-esteem in a specific context.
On another note, you suggest that an exploratory factor analysis (EFA) is required before conducting confirmatory factor analysis (CFA). We greatly appreciate this suggestion; however, we believe that it is not necessary to perform an exploratory factor analysis prior to confirmatory factor analysis in the context of evaluation from the perspective of Classical Test Theory (CTT) (Samejima, 1999) for several reasons:
-
Preexisting Theory: If the self-esteem instrument has already been validated in previous studies and is based on a well-defined theoretical structure, Confirmatory Factor Analysis (CFA) can be used directly to confirm that the data fit this pre-established structure. In this case, Exploratory Factor Analysis (EFA) would not provide any additional relevant information.
-
Focus on Validity: The Theory of Construct Validity (TCT) emphasizes the validity and reliability of the instrument. If there is a clear hypothesis about the internal structure of the test, CFA allows for evaluating whether the data support that hypothesis without needing to explore different possible structures through EFA.
-
Efficiency: Conducting an EFA can be a labor-intensive and time-consuming process, especially if there is already prior evidence regarding the structure of the instrument. By omitting this stage, resources are optimized and the validation process is accelerated.
-
Direct Confirmation: CFA allows for a direct assessment of how well the proposed model fits the observed data, providing fit indices that indicate whether the theoretical structure is valid. This is particularly useful when seeking to validate a specific instrument in a new population or context.
-
Simplicity in Interpretation: By avoiding EFA, the interpretative process is simplified by focusing on confirming a single theoretical structure, which facilitates communication of results and conclusions.
Consequently, if there is a solid theoretical foundation and prior evidence regarding the structure of the self-esteem instrument, the direct use of Confirmatory Factor Analysis (CFA) is sufficient to assess its validity and reliability without the need for a prior exploratory factor analysis.
In response to the suggestion related to section e), we would like to mention that although there is a previous study that has already validated the items for a university population, as indicated in the discussion, it is essential to translate the scale for the military population. This is because members of the Spanish Army face specific challenges, such as operational stress and discipline, which can influence their self-esteem. Therefore, a cultural and linguistic adaptation could ensure that the items are relevant and understandable for military personnel, allowing for more accurate data collection. Regarding your last suggestion, new substantial lines have been introduced that enhance the initial manuscript with the aim of appealing to a broad audience of readers.
Finally, we would like to thank you once again for your suggestions and the time invested, as they have contributed to improving the manuscript.

Round 2
Reviewer 3 Report
Comments and Suggestions for Authors
The manuscript “Validation of the Rosenberg Self-Esteem Scale in Military Personnel of the Spanish Army (RSE-JGA)” has been improved in some aspects but still needs improvements before publication.
1) Introduction:
a) As I pointed out in my previous review, the objective is clear, and it makes scientific sense to validate in a new context or with a specific sample, as is the case since it guarantees the scientific robustness of a measurement scale. However, other studies have previously adapted and validated the Rosenberg scale in Spanish people. The items on the scale are the same as those in this study. For example:
· Atienza, F. L., Moreno, Y. M., & Balaguer, I. S. (2000). Análisis de la dimensionalidad de la Escala de Autoestima de Rosenberg en una muestra de adolescentes valencianos. Revista de Psicología Universitas Tarraconensis, 22, 29-42.
· Martín-Albo, J., Núñez, J. L., Navarro, J. G., & Grijalvo, F. (2007). The Rosenberg Self-Esteem Scale: translation and validation in university students. The Spanish journal of psychology, 10(2), 458-467.
· Gómez-Lugo M, Espada JP, Morales A, Marchal-Bertrand L, Soler F, Vallejo-Medina P. (2016). Adaptation, Validation, Reliability and Factorial Equivalence of the Rosenberg Self-Esteem Scale in Colombian and Spanish Population. The Spanish Journal of Psychology. 2016;19:E66. doi:10.1017/sjp.2016.67
Therefore, no changes have been introduced to the items of the questionnaire, that is, the items have not been translated or adapted to the study sample. Thus, this study does not propose any advances in science.
b) The authors proposed 3 hypotheses. However, hypotheses 1 and 2 are not necessary because are the objective of the paper.
1) Materials and Methods:
a) In section 2.1. Participants and Procedures, you should explain why resilience was chosen to study validity. The authors describe it in section 2.2. Instruments, but it is more appropriate to include them in the procedure.
b) In the description of the instruments examples of items must be given when describing a scale.
c) As I pointed out in my previous review, section 2.4 is the most important question. Revision, translation and adaptation to the military context is inappropriate since the scale used already exists in Spanish. Nor has it been adapted because the items are the same. Then, this study does not propose any advance in science
2) Conclusions
This section does not discuss future studies but rather the most relevant contribution of this study.
Author Response
Dear reviewer, first of all thank you for your suggestions, we believe we have improved our work a lot. Here is a response to each of your suggestions:
1) Introduction:
Comment to you that this study provides an opportunity to corroborate the validity.
of Rosenberg scaling to the military context. As you know, the validation of a scaling cannot be generalized to a general population,
Therefore, this study confirms and underlines its validity, as well as its relationship with resilience, which in terms of morale, this variable is of utmost importance to achieve success in military operations.
b) Hypotheses 1 and 2 have been eliminated, so that the third hypothesis is presented as the only research hypotheses.
b) Hypotheses 1 and 2 have been eliminated, so that the third hypothesis is presented as the only research hypothesis.
2) Materials and methods:
(a) Section 2.1 explains why resilience was chosen to
study validity. 2.2 The instruments are included in the procedure section.
(b) Some of the items that make up the scale have been introduced as examples.
the scale.
c) This study proposes as an advance in science the reliability of the measurement of the level of self-esteem in military personnel and its relationship with the level of resilience.
2) Conclusions:
Suggestions for future studies have been eliminated, leaving only the contributions of this study.
